# A Novel Ultrasonographic Anthropometric-Independent Measurement of Median Nerve Swelling in Carpal Tunnel Syndrome: The “Nerve/Tendon Ratio” (NTR)

**DOI:** 10.3390/diagnostics12112621

**Published:** 2022-10-28

**Authors:** Paolo Falsetti, Edoardo Conticini, Caterina Baldi, Emilio D’Ignazio, Suhel Gabriele Al Khayyat, Marco Bardelli, Stefano Gentileschi, Roberto D’Alessandro, Miriana D’Alessandro, Caterina Acciai, Federica Ginanneschi, Luca Cantarini, Bruno Frediani

**Affiliations:** 1Rheumatology Unit, Department of Medical, Surgical Sciences and Neurosciences, University Hospital of Siena, 53100 Siena, Italy; 2Unit of Respiratory Diseases and Lung Transplantation, Department of Medical, Surgical Sciences and Neurosciences, University Hospital of Siena, 53100 Siena, Italy; 3Neurorehabilitation Unit, San Donato Hospital, 52100 Arezzo, Italy; 4Neurology Unit, Department of Medical, Surgical Sciences and Neurosciences, University Hospital of Siena, 53100 Siena, Italy

**Keywords:** carpal tunnel syndrome, median nerve, ultrasound, electrodiagnostics, neuropathy

## Abstract

Background: There is little consensus on ultrasound (US) normative values of cross-sectional area of median nerve (MN-CSA) in carpal tunnel syndrome (CTS) because of its dependency on anthropometric parameters. We aim to propose a novel anthropometric-independent US parameter: MN-CSA/flexor radialis carpi CSA (FCR-CSA) ratio (“Nerve Tendon Ratio”, NTR), in the diagnosis of clinically and electrodiagnostic (EDS)-defined CTS. Methods: 74 wrists of 49 patients with clinically defined CTS underwent EDS (scored by the 1–5 Padua Scale of electrophysiological severity, PS) and US of carpal tunnel with measurement of MN-CSA (at the carpal tunnel inlet), FCR-CSA (over scaphoid tubercle) and its ratio (NTR, expressed as a percentage). US normality values and intra-operator agreement were assessed in 33 healthy volunteers. Results: In controls, the mean MN-CSA was 5.81 mm^2^, NTR 64.2%. In 74 clinical CTS, the mean MN-CSA was 12.1 mm^2^, NTR 117%. In severe CTS (PS > 3), the mean MN-CSA was 15.9 mm^2^, NTR 148%. In CTS, both MN-CSA and NTR correlated with sensitive conduction velocity (SCV) (*p* < 0.001), distal motor latency (DML) (*p* < 0.001) and PS (*p* < 0.001), with a slight superiority of NTR vs. MN-CSA when controlled for height, wrist circumference and weight. In CTS filtered for anthropometric extremes, only NTR maintained a correlation with SCV (*p* = 0.023), DML (*p* = 0.016) and PS (*p* = 0.009). Diagnostic cut-offs were obtained with a binomial regression analysis. In those patients with a clinical diagnosis of CTS, the cut-off of MN-CSA (AUROC: 0.983) was 8 mm^2^ (9 mm^2^ with highest positive predictive value, PPV), while for NTR (AUROC: 0.987), the cut-off was 83% (100% with highest PPV). In patients with EDS findings of severe CTS (PS > 3), the MN-CSA (AUROC: 0.876) cut-off was 12.3 mm^2^ (15.3 mm^2^ with highest PPV), while for NTR (AUROC: 0.858) it was 116.2% (146.0% with highest PPV). Conclusions: NTR can be simply and quickly calculated, and it can be used in anthropometric extremes.

## 1. Introduction

Carpal tunnel syndrome (CTS) is the most common nerve entrapment disease. CTS is a clinical diagnosis, but electrodiagnostic study (EDS) and ultrasound (US) can confirm diagnosis. Even if there is no single gold standard test for confirming diagnosis, EDS is widely recommended in CTS as it can give a neurophysiological classification of MN neuropathy suggesting a surgical option [1]. On the other hand, US can provide information about anatomically and structurally concurrent conditions related to CTS [2], but it is not widely recommended for routine use [3]. Nevertheless, robust evidence suggests that US can accurately describe MN neuropathy, but there is not a complete consensus on the optimal US measurements in CTS and its normative values [4,5,6,7].

The commonest US measurement in CTS is the cross-sectional area of median nerve (MN-CSA) that describes the neuropathic swelling of the nerve at the inlet of carpal tunnel [1,2]. There is little consensus on normative values of MN-CSA because of its dependency on anthropometric parameters (overall wrist circumference and height). Consequently, there is a surprisingly high variability of diagnostic cut-off (varying between 8.5 mm and 15 mm) [2,4,5,6,8,9,10,11,12,13,14,15].

In fact, an earlier study showed a positive correlation between CSA and wrist circumference, height and weight [5]. For these reasons, smaller people have lesser wrist circumference and lesser CSA of the median nerve also in pathologic conditions. On the other hand, taller subjects can have MN-CSA over cut-off in the absence of compressive neuropathy (Figure 1) [16].

Consequently, various anthropometric-independent diagnostic US parameters have been proposed. In particular CSA modified by equations [11,17,18] and CSA ratios (wrist-to-forearm ratio, WFR) [19] are the most cited, but they should be confirmed in different populations, and they are complex and relatively time-consuming to perform in clinical practice.

In this context, we propose a simple and quick anthropometric-independent US parameter (median nerve-CSA/flexor carpi radialis-CSA ratio, expressed as a percentage, and abbreviated as nerve tendon ratio (NTR) throughout this text) in the diagnosis of clinically and EDS-defined CTS. In this paper, we aim to verify the dependency of NTR to anthropometric parameters with respect to crude MN-CSA and to compare the performance of NTR in comparison with MN-CSA in the diagnosis of clinical and EDS-confirmed CTS.

## 2. Materials and Methods

A total of 74 wrists of 49 unselected consecutive outpatients with clinically defined CTS underwent EDS in the Clinical Neurophysiology Unit from June 2021 to June 2022. The patients were referred by general practitioners or specialists because of upper limb symptoms in a context of real clinical practice.

The CTS clinical diagnosis was made according to American Academy of Neurology criteria [20] and to the consensus criteria of the classification of CTS [21], where only patients with paresthesia (numbness, tingling, burning) or pain in at least 2 of the first 3 fingers (classic/probable cases) were included in the study. Only idiopathic CTS were included in the study group, whereas secondary CTS were excluded. No patient had renal failure, connective tissue diseases, diabetes, hypothyroidism, or dysmetabolic polyneuropathies, or reported symptoms during pregnancy or lactation.

An EDS, including neurography of the median and ulnar nerves, was performed by the same neurophysiologist in line with the guidelines of the American Association of Electrodiagnostic Medicine [22]. The EDS was performed using a Synergy Medelec electromyography with surface recording electrodes. The details are reported in a previous study [23]. In brief, the considered MN electrophysiological parameters were distal motor latency (DML) calculated for a distance of 7 cm between wrist and abductor pollicis brevis (APB), compound muscle action potential amplitude (CMAP), orthodromic sensory conduction velocity (SCV) and sensory action potential (SAP) recorded in the third finger–wrist and fourth finger–wrist tracts.

The CTS was scored with the 1–5 Padua Scale (PS) of electrophysiological severity [24]. This scale evaluates the presence/absence of SAP and CMAP and normal/abnormal SCV and DML. 

On the same day of EDS, every patient underwent a high-resolution US examination (4–15 MHz and 8–24 MHz linear probes, MyLab X8 eXP Esaote and 6–18 MHz linear probe, MyLab Twice, Esaote, Genoa, Italy) of the carpal tunnel in the Rheumatology Unit of the same hospital. The examination was performed by a single rheumatologist with 20 years of experience in musculoskeletal US and blinded to clinical and electrophysiologic CTS severity. Patients were seated in a chair with arms extended, hands resting in a horizontal supine position and fingers relaxed. The median nerve cross-sectional area (CSA) was measured at the tunnel inlet (just before the proximal margin of the flexor retinaculum) by tracing the inner border of the thin hyperechoic rim of the nerve (perineurium) with the automatic ellipsoid technique (or manual tracing technique if the nerve had an irregular shape). Moreover, the CSA of flexor carpi radialis (FCR-CSA) was measured over (or just proximal to) the scaphoid tubercle, where the tendon runs parallel to the skin (possibly in the same scan of MN-CSA, with the automatic ellipsoid technique). The ratio between the two CSAs was calculated and recorded as the nerve tendon ratio (NTR) (expressed as a percentage) (Figure 2). The probe was applied without additional pressure, and the mean value of three measurements, along with clinical and anthropometric characteristics (weight, height, wrist circumference), was recorded.

To obtain control values of both CSAs and NTR, we enrolled 33 healthy volunteers without signs or symptoms of CTS. The weight and height of each participant were measured, as well as their wrist circumferences. Moreover, the US examination was performed in the dominant hand, measuring MN-CSA, FCR-CSA (mean values of three measurements) and calculating NTR normality values. Intra-operator agreement of US parameters was assessed in these healthy subjects.

The study was conducted in accordance with the tenets of the Declaration of Helsinki, and the use of clinical data for research purposes was approved by the local Ethics Committee of the University of Siena (Reference No. 22271, “RHELABUS”). Written informed consent was obtained for all the procedures according to the local Institutional review board guidelines.

## 3. Statistical Analysis

Data are expressed as mean ± standard deviation or numbers (%). For categorical variables, Fisher’s exact or Chi-squared tests were used to compare proportions between groups. Student’s t-test was used to compare the means of continuous variables between two groups when the distribution of data was normal, and with Welch’s correction otherwise.

A non-parametric Kruskal–Wallis test was used to compare the means of continuous variables among groups, while a Dwass–Steel–Critchlow–Fligner (DSCF) test was used for pairwise comparisons.

Non-parametric Spearman rank test was applied in order to correlate variables.

Multivariable linear regression was performed with all significant variables (wrist circumference, weight, gender, age, BMI, height, SCV, DML, PS) entered in a stepwise way to identify which factors independently correlated with the CSA, and this was checked for multicollinearity in both healthy and pathologic wrists.

Binomial logistic regression and receiver operating characteristic (ROC) curve analysis were used to determine the predictive diagnostic value of each US parameter in detecting CTS, with clinical diagnosis or EDS diagnosis used as a gold standard. The validity of each US parameter for diagnosis of CTS was determined by an estimation of the sensitivity and specificity of various cut-off points of both MN-CSA and NTR. Youden’s J-statistic method was applied to obtain the optimal cut-off values.

The coefficient of variation was calculated to estimate the precision of repeated measurements of CSAs.

Cohen’s k statistics were used to assess the agreement between the two alternative methods of categorical assessment (US and EDS in diagnosis of CTS) [25].

The level of statistical significance was set at a *p*-level of 0.05. Statistical analyses were performed using Jamovi and XLSTAT2021 statistical packages. 

The feasibility of SIJs PDUS/SWA was evaluated by recording the time spent by the operator and asking the patient about the comfortability of the examination.

## 4. Results

### 4.1. Participants Characteristics and Descriptive Statistics about MN-CSA and NTR in Healthy and Pathologic Wrists

Demographic, anthropometric, and clinical characteristics of participants and their comparisons are resumed in Table 1.

Among the 74 wrists with clinical CTS, EDS confirmed pathological NCVs in 71. Of these, 1 belonged to stage 1 (minimal CTS) of the electrophysiologic Padua severity scale, 24 to stage 2 (sensitive neuropathies), 27 to stage 3 (sensitive-motor neuropathies), 13 to stage 4 (absence of SAP), and 6 to stage 5 (absence of SAP and CMAP).

In healthy subjects, the mean values of MN-CSA were 5.81 mm^2^ (1.79 SD, range 4–10), 9.53 mm^2^ for FCR-CSA (1.79 SD, range 6–13) and 64.2% for NTR (14.3 SD, range 41.7–100%). In 74 wrists with clinical CTS, the mean values of MN-CSA were 12.1 mm^2^ (3.39 SD, range 6–22), 10.3 mm^2^ for FCR-CSA (1.95 SD, range 7–16) and 117% for NTR (29.1 SD, range 75–220) (Table 1). In wrists with severe CTS (PS > 3) the mean MN-CSA was 15.9 mm^2^ (3.82 SD, range 9.6–22), and the mean NTR was 148% (36.4 SD, range 100–220).

The coefficient of variation on repeated measures was 4.58% for MN-CSA and 3.9% for FCR-CSA in healthy wrists.

### 4.2. Demonstration of Independency from Anthropometric Parameters of NTR vs. MN-CSA

In healthy wrists, MN-CSA and FCR-CSA were strongly correlated to each other (r = 0.682, *p* < 0.001) and with height (r = 0.668, r = 0.732, respectively, *p* < 0.001), wrist circumference (r = 0.584, r = 0.556, respectively, *p* < 0.001) and weight (r = 0.448, *p* < 0.009 and r = 0.452, *p* < 0.008, respectively).

In pathologic wrists, both MN-CSA and NTR are well-correlated with SCV (r = −0.550, r = −0.611, respectively, *p* < 0.001), DML (r = 0.468, r = 0.558 respectively, *p* < 0.001) and PS (r = 0.636, r = 0.650, respectively, *p* < 0.001), with a slight superiority of NTR over MN-CSA when controlled for height, wrist circumference and weight (r = −0.623 vs. −0.558 for SCV, r = 0.606 vs. 0.470 for DML, and r = 0.674 vs. 0.642 for PS) (Appendix A).

In healthy wrists, a model of backward stepwise multivariate regression analysis showed that about 75% of the variability of MN-CSA (r^2^ = 0.754, *p* < 0.001) is dependent on height (*p* < 0.003), wrist circumference (*p* < 0.037) and weight (*p* < 0.016); while age, gender, and BMI are not independently correlated to CSA.

In pathologic wrists, a model of backward stepwise multivariate regression analysis showed that about half of the variability of MN-CSA is dependent on PS (r^2^ = 0.497, *p* < 0.001) as well as its derived NTR (r^2^ = 0.474, *p* < 0.001), while all anthropometric parameters such as height, wrist circumference, weight, age, gender and BMI are not independently determinant on MN-CSA or NTR, accounting for about 3% of its variability.

When the regression analysis was performed on wrists filtered for the extremes of anthropometric values (height and wrist circumference superior or inferior to 1 SD, i.e., height <156 cm or >175 cm and wrist circumference <15 cm or >18 cm), PS remained the major determinant of NTR variability (r^2^ = 0.323, *p* = 0.043 in height > 1 SD, r^2^ = 0.802, *p* < 0.001 in height < −1 SD, r^2^ = 0.544, *p* = 0.004 in wrist circumference > 1 SD, and r^2^ = 0.783, *p* = 0.003 in wrist circumference < −1 SD), while MN-CSA lacked significance in extremes of height. Similarly, the Pearson correlation analysis demonstrated that in wrists filtered for extreme anthropometric values, only NTR maintained a significant correlation with SCV (*p* = 0.023), DML (*p* = 0.016) and PS (*p* = 0.009).

### 4.3. Comparison of Diagnostic Performances of MN-CSA and NTR

In pathologic wrists, both US parameters had significant differences on ANOVA among various PS stages (*p* < 0.001), but both failed in pairwise DSCF comparisons between the first stages of PS (significant differences on mean values of both MN-CSA and NTR only between stages 2 vs. 4, 2 vs. 5, 3 vs. 5) (Appendix A).

In an ordinal regression model, both MN-CSA (*p* = 0.002) and NTR (*p* = 0.001) showed significant predictivity of PS stages in pathologic wrists (r^2^ = 0.306, *p* < 0.001), while all anthropometric parameters, such as height, wrist circumference, and weight, were not significantly determinant of PS stages.

A binomial regression analysis was performed to obtain diagnostic cut-offs from US parameters with different diagnostic gold standards (Appendix A).

A ROC curve was calculated considering each US parameter (MN-CSA and NTR) in the diagnosis of clinical CTS as the gold standard (Figure 3).

For MN-CSA, an estimated area under the curve (AUC) of ROC curve of 0.983 (SE 0.009, CI 95% 0.966–1) was obtained. The diagnostic cut-off obtained with Youden J-statistic was 8 mm^2^ (sensitivity 0.959, specificity 0.90). The cut-off with the best positive predictive value for clinical CTS diagnosis was 9 mm^2^ (sensitivity 0.838, specificity 1.00).

For NTR, an estimated area under the curve (AUC) of ROC curve of 0.987 (SE 0.01, CI 95% 0.968–1) was obtained. The diagnostic cut-off obtained with Youden J-statistic was 83.0% (sensitivity 0.946, specificity 0.967). The cut-off with the best positive predictive value for clinical CTS diagnosis was 100.0% (sensitivity 0.662, specificity 1.00).

Moreover, another ROC curve was calculated comparing each US parameter (MN-CSA and NTR) in the diagnosis of severe CTS (EDS-defined PS > 3) as the gold standard (Figure 4).

For MN-CSA, an estimated area under the curve (AUC) of ROC curve of 0.876 (SE 0.051, CI 95% 0.776–0.975) was obtained. The diagnostic cut-off obtained with Youden J-statistic was 12.3 mm^2^ (sensitivity 0.789, specificity 0.855). The cut-off with the best positive predictive value for EDS-confirmed PS > 3 CTS diagnosis was 15.3 mm^2^ (sensitivity 0.579, specificity 1.00).

For NTR, an estimated area under the curve (AUC) of ROC curve of 0.858 (SE 0.051, CI 95% 0.759–0.958) was obtained. The diagnostic cut-off obtained with Youden J-statistic was 116.2% (sensitivity 0.789, specificity 0.818). The cut-off with the best positive predictive value for EDS-confirmed PS > 3 CTS diagnosis was 146.0% (sensitivity 0.474, specificity 1.00).

Using the obtained US cut-off (MN-CSA 8 mm^2^, NTR 0.83), the Cohen statistic between the two diagnostic methods showed an overall concordance of 96.15%, with a fair agreement (k = 0.380) for EDS-defined diagnosis of CTS vs. MN-CSA, and an overall concordance of 92.30%, with fair agreement (k = 0.365) for the EDS-defined diagnosis of CTS vs. NTR.

A time of 20–30 s was sufficient for setting the US parameters and for completing the grey-scale US examination of MN and FCR with CSAs tracing and automatized calculation of NTR on a single wrist. No adverse events occurred during examinations, and all patients considered this examination quick, not painful, and mostly comfortable.

## 5. Discussion

There is a little consensus on US normal values of MN-CSA due to its dependency on anthropometric parameters (overall wrist circumference and height) with the consequent high variability of US diagnostic cut-offs (Table 2). Consequently, various anthropometric-independent diagnostic parameters have been proposed, but they are seldom used in clinical practice [11,17,18,19,26]. Only WFR is recommended in recent guidelines for studying particularly small or large nerves [1]. Our clinical observation of comparable CSAs between MN and FCR at the carpal tunnel inlet, derived from direct visual impression in the same US scan in healthy subjects, gave us the opportunity to study NTR as a potential US parameter of MN neuropathic swelling, regardless of anthropometric features.

Our study, to the best our knowledge, represents the first attempt to evaluate anthropometric-independent US features of MN, other than WFR, in the diagnostic work-up of CTS.

Our first impression has been confirmed in healthy subjects, where MN-CSA and FCR-CSA strongly correlate with each other and with anthropometric parameters. Moreover, we evidenced a good intra-observer reliability of NTR measurements in healthy subjects. NTR can be quickly calculated, often in the same US scan, with automatic function of ratio between ellipses or areas. All these characteristics have enabled us to test NTR as an anthropometric-independent measure of MN swelling.

Like MN-CSA, NTR also appeared significantly higher in patients affected by CTS than controls, and it appeared even higher in patients with advanced CTS.

As reported in previous studies [11] MN-CSA is strongly influenced by anthropometric factors in healthy subjects, but in pathologic wrists, the importance of anthropometric parameters is reduced as MN-CSA variability also depends on the severity of neuropathic swelling. Nevertheless, in conditions of anthropometric extremes, large normal nerves or small pathologic nerves could constitute a false positive or false negative in US examination. Even if both crude MN-CSA and NTR showed good correlations with SCV, DML, and finally with PS, in a generic population we demonstrated the superiority of NTR when patients were stratified according to anthropometric extremes. Particularly, in patients taller than 175 cm and shorter than 156 cm, NTR maintained an excellent correlation with PS, while such correlation was lacking for MN-CSA (Figure 5).

Moreover, both MN-CSA and NTR showed a probable limitation in the early stages of MN neuropathy, where the neuropathic swelling of the nerve is not so relevant. This aspect seems to be confirmed in our study, where the mean values of both MN-CSA and NTR did not show significant differences in the early stages of PS, whereas in later stages of PS, the measurements of MN-CSA (and its ratios) could be sufficient to describe the neuropathic swelling of the nerve. As is also suggested in recent studies, the early diagnosis of MN neuropathy in CTS probably requires a multi-parametric assessment [27,28]. This assessment should comprise an evaluation of MN echo structure and echo texture, MN stiffness (also in elasto-sonography and shear wave elastography), MN flattening ratio, and MN vascularity (also by novel microvascular imaging techniques) other than the bulging and thickness of the transverse carpal ligament [27,28,29,30,31].

Another aspect of our study is the production of different cut-off values of both US parameters for different clinical purposes. The indication of surgical treatment is strong for advanced stages (patients with persistent numbness or thenar atrophy), but there is little consensus for early and intermediate stages, where diagnostic studies such as EDS and US are often requested to confirm diagnosis and assess the severity of neuropathy [1,3,32]. Our study proposes cut-offs of both MN-CSA and NTR for severe MN neuropathy (PS > 3, that is conduction block of at least one action potential on EDS).

On the other hand, cut-offs for both US parameters are offered in the confirmation of the clinical suspicion of CTS-related MN neuropathy. These cut-offs are comparable with those proposed in previous studies, even if our MN-CSA cut-off for clinical diagnosis seems slightly lower with respect to studies from Northern European populations. This is probably due to significant anthropometric differences (in particular height) and different inclusion criteria and disease duration [4,11].

The revision of our CTS diagnosis with these obtained cut-offs, confirms a fair diagnostic concordance between EDS and US, as suggested in previous studies [6].

A limitation of our study is the lack of inter-observer agreement; we offer only satisfactory data on intra-observer agreement in healthy volunteers. This is because the pathologic cases were enrolled during the routine clinical practice for outpatients, and we could not perform a prolonged examination with multiple observers.

Other limitations of this study are the small population sampled and the unequal sizes of the studied and control groups. Even if NTR seems to offer good diagnostic performance in anthropometric extremes, further studies on larger samples and in different institutions and populations are needed for confirmation.

Finally, an intrinsic limitation of our study is the need for an experienced operator trained in musculoskeletal disorders, indeed, the proper assessment of FCR can be difficult and requires an optimal placement of the probe to avoid anisotropy. Moreover, FCR-CSA may be increased in the cases of tenosynovitis or tendinosis, which were not evidenced in any of our patients.

In conclusion, NTR is a novel anthropometric-independent US measurement of MN neuropathy in CTS. NTR seems to offer diagnostic performances comparable to crude MN-CSA but is less influenced by anthropometric parameters. Moreover, NTR can be simply and quickly calculated, and cut-off values for the diagnostic confirmation of both clinical CTS and severe CTS have been proposed. NTR could be used after crude MN-CSA calculation in anthropometric extremes.

## Figures and Tables

**Figure 1 diagnostics-12-02621-f001:**
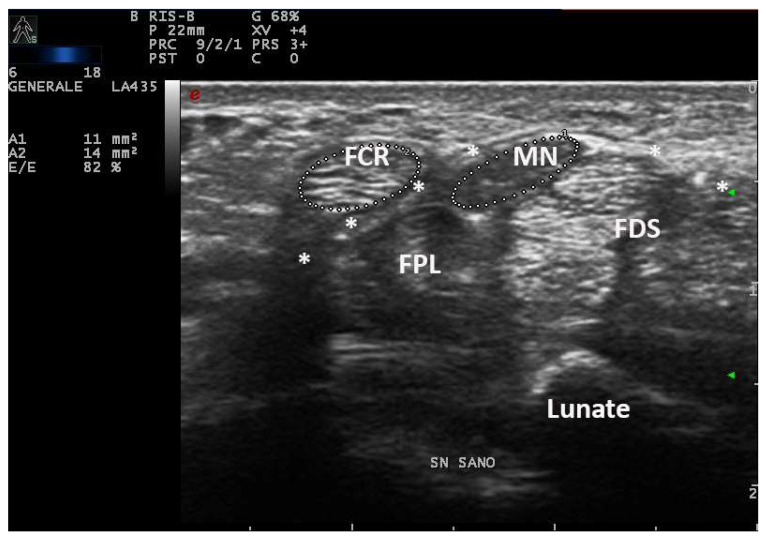
Transverse scan over left carpal tunnel of a tall male, asymptomatic for CTS in left hand (180 cm × 85 kg, 19 cm of wrist circumference). Linear 6–18 MHz probe. The MN-CSA (area 1) results of 11 mm^2^ are suggestive of pathologic swelling, but EDS was normal (PS = 0, performed as control side, for CTS in the right side). The flexor carpi radialis (FCR) shows a CSA (area 2) of 14 mm^2^. MN = median nerve, FPL = flexor pollicis longus, * = transverse ligament, FDS = flexor digitorum superficialis.

**Figure 2 diagnostics-12-02621-f002:**
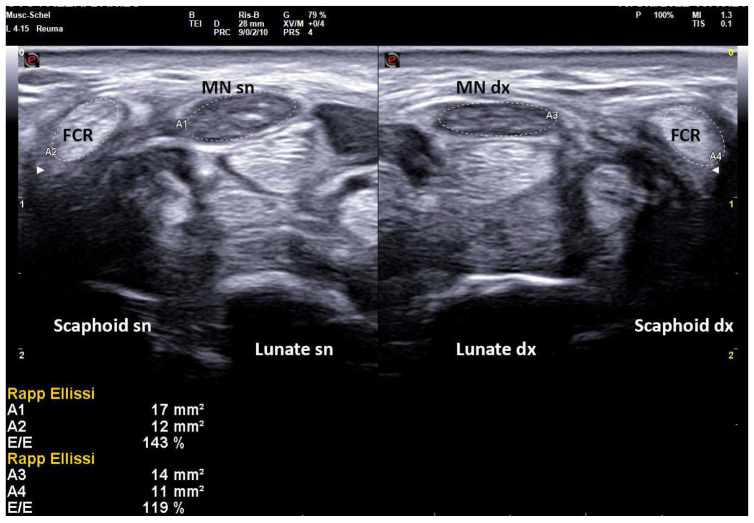
Transverse scan over left and right carpal tunnels in male patient (172 cm × 82 kg, 18.5 cm of wrist circumference) affected with extremely severe CTS (Padua Scale 5 bilaterally). Linear 4–15 MHz probe. Both MN-CSA show frankly pathologic values (Area 1, left: 17 mm^2^, Area 3, right: 14 mm^2^). NTR is calculated as the ratio between MN-CSA and FCR-CSA (expressed as a percentage), and shows bilaterally pathologic values (left 143%, right 119%) indicative for severe CTS. MN = median nerve, FCR = flexor carpi radialis.

**Figure 3 diagnostics-12-02621-f003:**
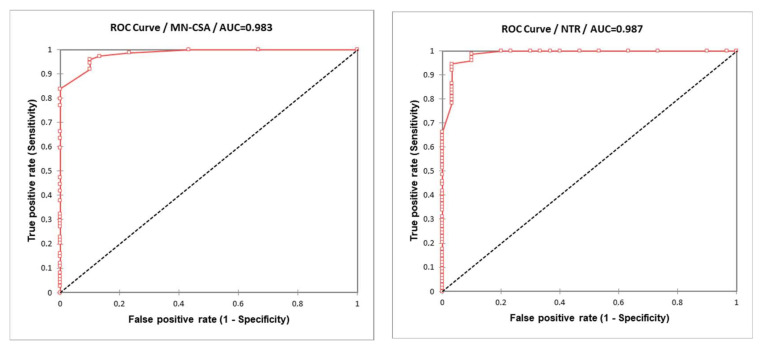
ROC curves obtained comparing each US parameter (MN-CSA and NTR) in the clinical diagnosis of CTS (as gold standard).

**Figure 4 diagnostics-12-02621-f004:**
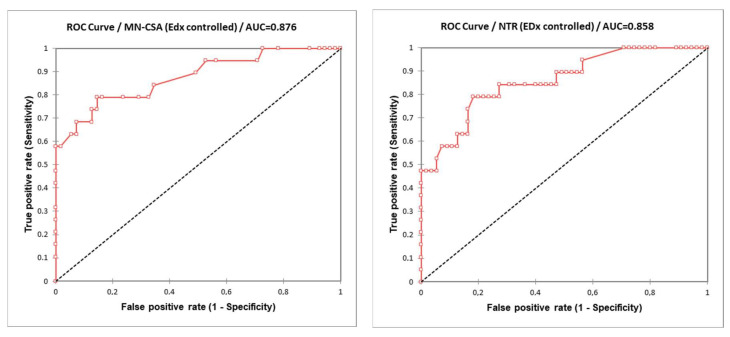
ROC curves obtained comparing each US parameter (MN-CSA and NTR) in severe EDS-defined (PS > 3) diagnosis of CTS (as gold standard).

**Figure 5 diagnostics-12-02621-f005:**
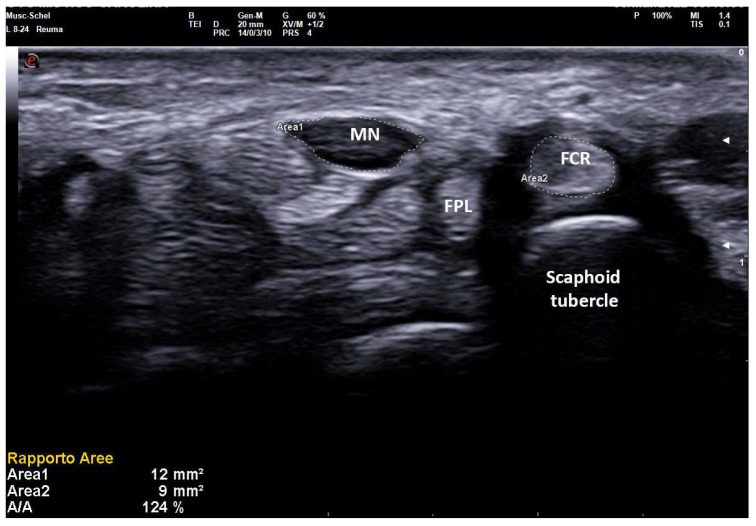
Transverse scan over right carpal tunnel in a small female patient (150 cm × 50 kg, 13.7 cm of wrist circumference) affected with medium-grade CTS (Padua Scale 3, SCV 37.1 m/sec, DML 4.8 ms). Linear 8–24 MHz probe. Both areas of median nerve (MN) and flexor carpi radialis (FCR) were manually traced. A MN-CSA of 12 mm^2^ is not always considered pathologic (cut-off for MN neuropathy until 14–15 mm^2^ in some studies on CTS) and the severity is often underestimated because of the anthropometric characteristics of the patient. The NTR shows values unequivocally pathologic (124%), as MN-CSA is compared with an anatomic structure (FCR-CSA) that maintains small dimensions in CTS, following the anthropometric characteristics of the patient.

**Table 1 diagnostics-12-02621-t001:** Demographic, anthropometric, and clinical characteristics of patients and controls.

Subjects Characteristics	CTS Patients	Healthy Controls	Statistical Significance
Number of subjects (female/male)	49 (31/18)	33 (17/16)	*p* = 0.255 n.s.
Age, in years (±SD, range)	64.0 (±12.7, 31–85)	58.6 (±15.9, 24–77)	*p* = 0.092 n.s.
Height, in cm (±SD)	166.3 (±9.32)	170.03 (±9.6)	*p* = 0.087 n.s.
Weight, in kg (±SD)	73.4 (±16.48)	66.93 (±13.14)	*p* = 0.052 n.s.
Wrist circumference, in cm (±SD)	16.8 (±1.43)	16.03 (±1.98)	*p* = 0.06 n.s.
Disease duration, in months (±SD)	29.2 (±25.8)	n.a.	n.a.
MN-CSA, in mm^2^ (±SD, range)	12.1 (±3.39, 6–22)	5.81(±1.79, 4–10)	*p* < 0.0001 **
FCR-CSA, in mm^2^ (±SD, range)	10.3 (±1.95, 7–16)	9.53 (±1.79, 6–13)	*p* = 0.073 n.s.
NRT, in % (±SD, range)	117 (±29.1, 75–220)	64.2 (±14.3, 41.7–100)	*p* < 0.0001 **

Data are expressed as mean (±standard deviation, SD), if not otherwise specified. The level of statistical significance was set at a *p*-level of 0.05. n.s. = not significant, ** = *p* < 0.01, n.a. = not assessed. CTS = carpal tunnel syndrome, SD = standard deviation, MN-CSA = median nerve cross-sectional area, FCR-CSA = flexor carpi radialis cross-sectional area, NTR = nerve/tendon ratio.

**Table 2 diagnostics-12-02621-t002:** MN-CSA and NTR in healthy patients and CTS.

MN at Inlet: US Parameters	Present Study	Summary of Literature Findings
Mean MN-CSA in healthy (range)	5.81 mm^2^ (4–10 mm^2^)	7–9.33 mm^2^ [4,10]
Mean MN-CSA in CTS (range)	12.1 mm^2^ (6–22 mm^2^)	8–16.47 mm^2^ [4,8,10]
Cut-off MN-CSA for CTS diagnosis highest PPV	8 mm^2^ (sens. 0.959, spec. 0.90) 9 mm^2^ (sens. 0.838, spec. 1.00).	8.5–15 mm^2^ (sens. 0.78% ± 6, spec. 0.77 ± 6) [4,6,10,15]
Cut-off NTR for CTS diagnosis highest PPV	83.0% (sens. 0.946, spec. 0.967) 100.0% (sens. 0.662, spec. 1.00).	n.a. n.a.
Cut-off NTR for severe CTS (PS > 3) highest PPV	116.2% (sens. 0.789, spec. 0.818) 146.0% (sens. 0.474, spec. 1.00).	n.a. n.a.

The values of median nerve cross-sectional area (MN-CSA) (expressed in mm^2^) and nerve/tendon ratio (NTR) (expressed as a percentage) obtained in our population were reported. A brief summary of literature findings on this topic is also reported. US = ultrasound, CTS = carpal tunnel syndrome, sens. = sentivity, spec. = specificity, PPV = predictive positive value, n.a. = not assessed.

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
