# Peer review of "A Novel Ultrasonographic Anthropometric-Independent Measurement of Median Nerve Swelling in Carpal Tunnel Syndrome: The “Nerve/Tendon Ratio” (NTR)"

_diagnostics, 2022, doi:10.3390/diagnostics12112621_

Round 1

Reviewer 1 Report

This is an interesting manuscript proposing a new tool in ultrasound diagnostics of CTS. The value of the study is limited by a small patient group. I have several comments:

  1. The widely recognized name of the muscle is flexor carpi radialis with the abbreviation FCR, the same applies to flexor pollicis longus (FPL).

  2. Abstract, line 7 from the bottom: CTS should be written instead of STC.

  3. The unequal sizes of the studied and control groups (74 vs. 33 wrists) should be included as one of the limitations of the study.

  4. Inclusion and exclusion criteria should be clearly listed in the Materials and Methods section.

  5. Did any of the patients suffer from hypothyroidism?

  6. The indication for surgical treatment is strong for advanced stages, but there is little consensus in intermediate stages, where diagnostic studies as EDS and US are often requested to deepen the severity of neuropathy.” This sentence needs re-writing for a clear meaning.

  7. When discussing other ratios used in ultrasound diagnostics of CTS, I suggest including the IOR ratio as presented in the article by T. Fu, M. Cao, F Liu et al. entitled „Carpal Tunnel Syndrome Assessment with Ultrasonography: Value of Inlet-to-Outlet Median Nerve Area Ratio in Patients versus Healthy Volunteers” published in 2015.

  8. What are, in the Authors' opinion, the limitations of the proposed method?

  9. The manuscript requires assessment by a statistical editor

  10. The manuscript requires English language proofreading and editing.

Reviewer 2 Report

Thanks to the authors for this well-designed and pertinent study that defines a new ultrasound obtained ratio to better diagnose CTS, especially in patients in anthropometric extremes. 

Here are my suggestions to improve the manuscript.

Abstract:

·      You use “STC” as an abbreviation, but I think this was a typo.

·      Non grammatical sentence: “In severe EDS-defined CTS (PS>3): MN-CSA (AUROC: 0.876): 12.3 mm2 (15.3 mm2 37 with highest PPV), NTR (AUROC: 0.858): 116.2% (146.0% with highest PPV).”

Intro:

·      “but it is not widely recommended for routinely use” –> for ROUTINE use.

·      “There is a little consensus” ïƒ There is little consensus 

·      Please reword this sentence or make it into 2 sentences for improved readability: “For these reason smaller people have lesser wrist circumference and lesser CSA of the median nerve, also in pathologic condition, whereas taller subjects can have MN-CSA over cut-off in absence of compressive neuropathy (Figure 1) [16].”

·      Percentual ïƒ  percentage

·      Remove “lesser” in “lesser dependency” 

Materials & methods

·      “20-years-experienced” ïƒ  with 20 years of experience in musculoskeletal ultrasound 

·      Were the height & weight measured for each participant, or reported height / weight used?

Figure 2

·      I would remove this part of the sentence: “indicative for CTS with surgical indication” as this study has not been designed to evaluated NTR to predict surgical candidates or surgical response 

Results

·      “In pathologic wrists both MN-CSA and NTR well correlated” ïƒ  In pathologic wrists both MN-CSA and NTR ARE well correlated 

·      Non grammatical sentence: “For MN-CSA, an estimated area under the curve (AUC) of ROC curve of 0.983 (SE 234 0.009, CI 95% 0.966-1).”

Discussion

·      Missing word: “As reported in previous studies [11] MN-CSA is strongly influenced by anthropometric factors in healthy” SUBJECTS/WRISTS

·      Typo: NM ïƒ  MN

·      This article (Lin, Chih-Peng, et al. "Utility of ultrasound elastography in evaluation of carpal tunnel syndrome: a systematic review and meta-analysis." Ultrasound in medicine & biology 45.11 (2019): 2855-2865.) could be a pertinent reference to this sentence (The multi-parametric evaluation should comprise evaluation of MN echo-structure and echo- texture, MN stiffness (also in elasto-sonography and shear wave elastography), MN flattening ratio, MN vascularity (also by novel microvascular imaging techniques) other than 314 bulging and thickness of transverse carpal ligament 

·      It is not clear what the authors mean by “deepen the severity of neuropathy”

·      I would not keep that sentence “Our study proposes cut-offs of both MN-CSA and NTR for severe MN neuropathy (PS>3, 320 that is conduction block of at least one action potential on EDS) that constitute an unequivocal indication to surgery in the clinical practice. » as the study was not designed to answer that research question

·      It would really add to the reader’s experience to have in the discussion a summary table of the previously proposed US values for MN neuropathy at the wrist (a visually appealing table that could be used to compare the validated values in the literature).

Conclusion:

·      CTS patients with anthropometric extremes. ïƒ  IN anthropometric extremes 
